# Dynamic Partition Gaussian Crack Detection Algorithm Based on Projection Curve Distribution

**DOI:** 10.3390/s20143973

**Published:** 2020-07-17

**Authors:** Dan Xue, Weiqi Yuan

**Affiliations:** Computer Vision Group, School of Electronic and Information Engineering, Key Laboratory of Machine Vision, Shenyang University of Technology, Shenyang 110870, China; xued@sut.edu.cn

**Keywords:** tunnel crack detection, dynamic partitioned Gaussian, gray projection curve distribution, uneven illumination

## Abstract

When detecting the cracks in the tunnel lining image, due to uneven illumination, there are generally differences in brightness and contrast between the cracked pixels and the surrounding background pixels as well as differences in the widths of the cracked pixels, which bring difficulty in detecting and extracting cracks. Therefore, this paper proposes a dynamic partitioned Gaussian crack detection algorithm based on the projection curve distribution. First, according to the distribution of the image projection curve, the background pixels are dynamically partitioned. Second, a new dynamic partitioned Gaussian (DPG) model was established, and the set rules of partition boundary conditions, partition number, and partition corresponding threshold were defined. Then, the threshold and multi-scale Gaussian factors corresponding to different crack widths were substituted into the Gaussian model to detect cracks. Finally, crack morphology and the breakpoint connection algorithm were combined to complete the crack extraction. The algorithm was tested on the lining gallery captured on the site of the Tang-Ling-Shan Tunnel in Liaoning Province, China. The optimal parameters in the algorithm were estimated through the Recall, Precision, and Time curves. From two aspects of qualitative and quantitative analysis, the experimental results demonstrate that this algorithm could effectively eliminate the effect of uneven illumination on crack detection. After detection, Recall could reach more than 96%, and after extraction, Precision was increased by more than 70%.

## 1. Introduction

The highway maintenance center is mandated to regularly inspect the tunnel lining to make timely repairs and ensure the safety of the tunnel. In recent years, in tunnel lining inspection, visual inspection technology has been introduced to replace traditional manual inspection, and the efficiency has been greatly improved. Various countries have invented tunnel lining inspection vehicles and installed image acquisition devices on the vehicles to perform defect detection on the captured lining images using image processing algorithms. In addition to the camera, the acquisition device needs to add additional light source equipment to ensure the quality of the picture being captured. Although an external light source can ensure that the image captured by the vehicles for high speed is clear, it cannot guarantee the uniformity of the image illumination. When detecting defects under uneven illumination conditions, if the same threshold detection is used for strong and weak illumination, it will inevitably lead to missed detection and false detection. Moreover, this phenomenon also occurs when cracks are detected in images captured in the areas of underwater dams [1,2,3], highway pavements [4,5,6], and bridges [7,8,9]. When the width and number of cracks exceed the allowable range, it will lead to structural decay and affect compressive strength variation [10], structural response, and seismic fragility [11]. It should cause high attention, be detected, and repaired in a fast amount of time.

There are two main research methods for crack detection on images of uneven illumination. One method is to equalize the image first and then detect the cracks globally. For example, in References [12,13,14,15,16,17,18,19,20], low-hat transform, mask uniformity algorithm, the Hessian matrix to construct a linear filter, Retinex, background difference method, etc., were used to balance the image illumination and then detect cracks. Although the image equalization process can change the overall brightness distribution of the image, the contrast to the cracks at the darker areas increased, and in the lighter areas, decreased. But even one crack, it had some places that were deep and shallow, so there was still no guarantee that the crack would be detected completely with the same threshold.

Another method is to directly use the method of the multi-scale space and set a local threshold to detect cracks. For example, in Reference [21], a crack detection algorithm based on sub-regional multi-scale analysis was proposed to extract the grayscale, entropy, and texture feature parameters of the crack and its surroundings from different scales of the image to obtain information such as the direction trend and bending degree of the crack on different scales. However, for images with uneven illumination, these extracted features will be affected and which will affect the final detection results. Reference [22] proposed a crack detection method based on fractional Fourier transforms. Fractional-order Fourier transforms from different orders correspond to different time–frequency domains to remove stains and extract cracks. However, the automatic selection of fractional orders is not implemented. Reference [23] proposed an automatic recognition method of cracks at tunnel linings based on local grid features of images. But this method does not have a good effect on the image which has no obvious differences between brightness and contrast with cracked and the surrounding background pixels. Reference [24] used probability maps based on the multi-scale neighborhood information to detect cracks. However, if the brightness and contrast differences between crack and the surrounding background pixels were not obvious, it was impossible to obtain obvious probability differences and caused missed detection.

In recent years, relevant scholars have researched the application of deep learning in the detection of cracks at tunnel linings. References [25,26,27] achieved good results in terms of accuracy and efficiency. Nevertheless, deep learning algorithms require a large number of labeled images of training, and these images are not easy to obtain.

Therefore, when the brightness and contrast to the cracked pixels and the surrounding background pixels in the image of uneven illumination are inconsistent, this paper proposes a multi-scale detection method that does not require lighting equalization and can dynamically set the local threshold based on the distribution of the projection curve. In this method, the detection area is divided by the distribution of the projection curve; the dynamic projection background gray value is introduced to improve the set of the threshold of the original Gaussian model; the relationship between the Gaussian scale factor and the line width is retained. Therefore, a dynamic partition Gaussian (DPG) model of the projected background gray value, gradient value, and Gaussian scale factor is established to complete multi-scale crack detection. In the process of crack extraction, the two features of the maximum ellipse diameter and the curve degree of the line are combined to eliminate pseudo cracks and connect the real crack breakpoints.

The remainder of the paper is organized as follows. The feature of detection objects is analyzed in Section 2. Section 3 introduces the multi-scale Gaussian model. Section 4, Section 5 and Section 6 discussed the proposed improved model (DPG), crack extraction, and algorithm steps in this paper, respectively. We analyze and evaluate the performance of the proposed algorithm in the lining gallery captured at the site of Tang-Ling-Shan Tunnel in Liaoning Province, China, in Section 7, and Section 8 concludes the paper.

## 2. Feature Analysis of Detection Objects

In the lining gallery captured att the site of Tang-Ling-Shan Tunnel in Liaoning Province, China, an image is selected to observe the distribution of its horizontal (vertical) grayscale projection curves, as shown in Figure 1.

The gray value distribution of the image in Figure 1 is described by the black horizontal projection curve and the red vertical projection curve. Where, the horizontal axis of the horizontal (vertical) projection curve represents the column (row) serial number of the image, and the vertical axis represents the gray mean of the image counted by columns (rows). Therefore, the gray value of the vertical axis can reflect the brightness and darkness of the image, that is, the distribution of light. So, it can be seen that the overall image is dark, and the light distribution is uneven, and the middle is brighter than both sides. The lines of the image appear as abrupt changes in the projection curve. The more obvious the difference between the line and the surrounding background, the more obvious the abrupt the features. For example, the abrupt changes of lightboxes (marks 2, 3) and brick seams (marks 1, 4) can be reflected through the gray projection curve (vertical projection in the blue box and horizontal projection in the red box). However, because the difference between the crack and the surrounding background is not obvious, the abrupt change characteristics in the projection curve are weak (marked by a thick red frame 5) and affected by the uneven illumination, when the cracks are in different positions in an image, the difference between the cracks and the surrounding background is still inconsistent. Therefore, it is necessary to use a large threshold when the difference is large and a small threshold when the difference is small. For a darker image, where the light is stronger, the difference degree is larger, and the projection curve can well reflect the distribution of light intensity.

The algorithm in this paper includes two parts. In the first part, it is proposed a DPG model to detect cracks, and, in the second part, it comprises morphology and a breakpoint connection algorithm to extract cracks. The framework of the algorithm is shown in Figure 2.

## 3. Multi-Scale Gaussian Model

The scale-space L(x,y,σ) of an image is defined as the convolution operation of the original image I(x,y) with a variable-scale 2D Gaussian function G(x,y,σ).

For example, the scale-space form is expressed as:(1)L(x,y,σ)=G(x,y,σ)*I(x,y)

Among them is,
(2)$$G\left(x,y,\sigma \right)=\frac{1}{2{\pi \sigma }^{2}}e^{-\frac{x^{2}+y^{2}}{2\sigma^{2}}}$$
where (x,y) is the spatial coordinate, and σ is the scale factor whose size determines the smoothness of the image. Multiple Gaussian scale-spaces can be constructed by selecting different scale factors σ.

The crack lines have the feature that the weight of the center point is the largest and decreases gradually to the four directions which is very similar to the Gaussian distribution, so the Gaussian model can be used to detect the lines. The feature extraction of the lines can be determined by three parameters of the width and direction of lines and the gradient perpendicular to lines.

### 3.1. Relationship between Line Width and the Gaussian Model

Reference [28] has shown that σ has a relationship of σ≥w/2√3 with the line width w. As the larger σ, the more the image is smoothed, and the more detrimental it is to edge detection. So, choose the minimum, i.e.:(3)$$\sigma =\frac{w}{2\surd 3}$$

By changing σ, lines of different widths can be detected. 

### 3.2. Relationship between the Direction and Gradient of the Line and Gaussian Model

At a certain scale, when a Gaussian model is used to detect lines, the partial derivatives of $r_{x}$, $r_{y}$, $r_{xx}$, $r_{yy}$, and $r_{xy}$ of image I(x,y) can be estimated. This partial derivative can be realized by convolving the image I(x,y) with the derivative of the Gaussian smoothing kernel. The Gaussian convolution kernel is as follows:(4)$$\{\begin{array}{c} G_{x}\left(x,y,\sigma \right)=\frac{\partial G\left(x,y,\sigma \right)}{\partial x}\cdot G\left(x,y,\sigma \right)\\ G_{y}\left(x,y,\sigma \right)=\frac{\partial G\left(x,y,\sigma \right)}{\partial y}\cdot G\left(x,y,\sigma \right)\\ G_{xx}\left(x,y,\sigma \right)=\frac{\partial^{2}G\left(x,y,\sigma \right)}{\partial x^{2}}\cdot G\left(x,y,\sigma \right)\\ G_{yy}\left(x,y,\sigma \right)=\frac{\partial^{2}G\left(x,y,\sigma \right)}{\partial y^{2}}\cdot G\left(x,y,\sigma \right)\\ G_{xy}\left(x,y,\sigma \right)=\frac{\partial G\left(x,y,\sigma \right)}{\partial y}\cdot \frac{\partial G\left(x,y,\sigma \right)}{\partial x}\cdot G\left(x,y,\sigma \right) \end{array}$$
(5)$$\{\begin{array}{c} r_{x}=I\left(x,y\right)*G_{x}\left(x,y,\sigma \right)\\ r_{y}=I\left(x,y\right)*G_{y}\left(x,y,\sigma \right)\\ r_{xx}=I\left(x,y\right)*G_{xx}\left(x,y,\sigma \right)\\ r_{yy}=I\left(x,y\right)*G_{yy}\left(x,y,\sigma \right)\\ r_{xy}=I\left(x,y\right)*G_{xy}\left(x,y,\sigma \right) \end{array}$$

The direction and gradient of the line can be determined by calculating the eigenvalues and eigenvectors of Hessian matrix H(x,y).
(6)$$H\left(x,y\right)=\left(\begin{array}{c} r_{xx}\;r_{xy}\\ r_{xy}\;r_{yy} \end{array}\right)$$

The Hessian matrix has two eigenvalues and two corresponding eigenvectors. The two eigenvalues indicate the anisotropy of the image change in the directions pointed by the two eigenvectors. The stronger the linearity is, the more anisotropic are. Therefore, for the line to be detected, the maximum absolute eigenvalue of the Hessian matrix corresponds to the second-order gradient value in the normal direction, and the feature vector corresponds to the direction of normal $\left(n_{x},n_{y}\right)$.

Among them,
(7)$${\left|\left|\left(n_{x},n_{y}\right)\right|\right|}_{2}=1$$
(8)$$t=\frac{r_{x}n_{x}+r_{y}n_{y}}{r_{xx}n_{x}^{2}+2r_{xy}n_{x}n_{y}+r_{yy}n_{y}^{2}}$$

When$\;\left(tn_{x},tn_{y}\right)\in \left(0.5,0.5\right)\times \left(-0.5,0.5\right)$, i.e., the first-order zero-crossing point of the edge is in the current pixel, and the second derivative of second-order gradient values and the direction of the normal indicates the strength of the line. When the strength value is greater than the threshold *T*, the point is the center point of the line to be detected.

## 4. Proposed Improved Model

The crack appears as a low gray value in the image. When the crack is on a background with higher brightness, the gradient value is larger. In the tunnel lining image, the pixels occupied by cracks are much smaller than the background pixels, then the background gray value can be reflected by the gray value of the projection curve. Therefore, the greater the gray value of the projection curve, the greater the gradient value of the crack line. Then, it can be seen that in the original Gaussian model, the threshold *T* is related to the gradient of the line, and this gradient is related to the gray value of the projection curve. Therefore, a new Gaussian model threshold *T* is constructed:(9)T(t)=z(t)+g(t)

Where, T(t) is the dynamic local threshold of the line to be detected, z(t) is the projected background gray value near the line, and g(t) is the minimum gradient value of the line.

Therefore, this paper proposes a DPG model. Among them, a scale factor σ and threshold *T* are the keys of the algorithm. When using this model for line detection, the area is divided by the image projection curve, and a dynamic local threshold *T* was set to detect the center point of the line. Different scale factors σ can be set to detect lines of different widths.

### 4.1. Image Gray Projection Curve

If the size of an image I(x,y) is M×N, then the gray value of the image is subjected to the cumulative average projection in the vertical (horizontal) direction to obtain a vertical (horizontal) gray projection function PV(x)
(PH(y)). The curve drawn is the image gray projection curve.
(10)$$PV\left(x\right)=\frac{\sum_{y=1}^{N}z\left(x,y\right)}{N}$$
(11)$$PH\left(y\right)=\frac{\sum_{x=1}^{M}z\left(x,y\right)}{M}$$

### 4.2. Dynamic Partition Division Rules

The gray projection function reflects the intensity distribution trend of an image. When the same object is detected in an image, the threshold size is set according to the size trend of the projection function., i.e., the larger the projection function value is, the larger the threshold value needs to be set, and conversely, the smaller. Therefore, the region can be divided dynamically according to the projection curve, and the dynamic local threshold T is set, the definition rules are as follows:

(1) Boundary condition

To ensure that cracks can be detected in the darkest part of the image background, the minimum gray value $V_{1}\left(H_{1}\right)$ in the vertical (horizontal) projection curve is taken as the boundary condition of the background gray value z(t) partition.
(12)$$V_{1}=min\left(PV\left(x\right)\right)$$
(13)$$H_{1}=min\left(PH\left(y\right)\right)$$

(2) The number of partitions

The difference values between the maximum and minimum gray values in the projection curve are divided by the partition boundary conditions and then rounded to the number of z(t) partitions. The number of vertical projection partitions is:(14)$$n_{V}=n_{Vlt}+1+n_{Vrp}=\left(\frac{max\left(PV\left(x\right)\right)-PV\left(0\right)}{V_{1}}\right)+1+\left(\frac{max\left(PV\left(x\right)\right)-PV\left(M\right)}{V_{1}}\right)$$

The number of horizontal projection partitions is:(15)$$n_{H}=n_{Hut}+1+n_{Hdp}=\left(\frac{max\left(PH\left(x\right)\right)-PH\left(0\right)}{H_{1}}\right)+1+\left(\frac{max\left(PH\left(x\right)\right)-PH\left(N\right)}{H_{1}}\right)$$

(3) Threshold *T*

According to the gray value of the projection curve, the regions can be divided into the left half, the middle half, and the right half as shown in Equations (14–15). Different thresholds are set in different partitions, and the vertical threshold $T_{V}$ and horizontal threshold $T_{H}$ are obtained.
(16)$$T_{V}=\{\begin{array}{c} \left(PV\left(0\right)+V_{1}\cdot n_{l}\right)-k\cdot V_{1},\;\;\;\;0\leq n_{l}\leq n_{Vlt}-1\\ min\left(\left(PV\left(0\right)+V_{1}\cdot n_{Vlt}\right),\left(PV\left(M\right)+V_{1}\cdot n_{Vrp}\right)\right)\\ \left(PV\left(M\right)+V_{1}\cdot n_{r}\right)-k\cdot V_{1},\;\;\;\;0\leq n_{r}\leq n_{Vrp}-1 \end{array}-k\cdot V_{1}$$
(17)$$T_{H}=\{\begin{array}{c} \left(PH\left(0\right)+H_{1}\cdot n_{u}\right)-k\cdot H_{1},\;\;\;\;0\leq n_{u}\leq n_{Hut}-1\\ min\left(\left(PH\left(0\right)+H_{1}\cdot n_{Hut}\right),\left(PH\left(N\right)+H_{1}\cdot n_{Hdp}\right)\right)\\ \left(PH\left(N\right)+H_{1}\cdot n_{d}\right)-k\cdot H_{1},\;\;\;\;0\leq n_{d}\leq n_{Hdp}-1 \end{array}-k\cdot H_{1}$$

Where the value of *k* is related to the weakest point between the object to be detected and the surrounding background. The weaker, the larger the *k* needs to be set. On the contrary, the smaller *k* is.

### 4.3. Multi-Scale Gaussian Crack Detection

Different scale factors σ can detect cracks with different widths. Observe the cross-sections of the lines with various widths in Figure 1, as shown in Figure 3. 

In Figure 3, the difference between crack and pseudo crack widths is analyzed. It can be seen that no matter morphology, gray value, or gradient value, they are very similar and difficult to separate. But the width of pseudo cracks (e.g., light box edges, brick seams,) is mostly wider than the cracks. Therefore, set the appropriate width value and substitute it into Equation (3) to get different scale factors σ.

## 5. Crack Extraction

Because of the discontinuity of the cracks in the lining (i.e. a macro view is a crack and the micro-view are several small lines), the morphological characteristics of the crack are used to extract the complete crack.

(1) First, the diameter of the largest circumscribed ellipse of the line is used as the coarse screening condition, and the longer line detected by DPG is extracted as the mainline of the suspected crack.
(18)$$D_{M}=\sqrt{{\left(y_{2}-y_{1}\right)}^{2}+{\left(x_{2}-x_{1}\right)}^{2}}.$$

In Figure 4, $\left(x_{1},y_{1}\right)$ and $\left(x_{2},y_{2}\right)$ are the end coordinates of the line, and the maximum diameter of the circumscribed ellipse is represented by $D_{M}$. The lengths of most cracks are longer than pseudo cracks. Therefore, when $D_{M}\geq T_{M}$, it is judged as the mainline of the suspected crack.

(2) Then, it is connected to the endpoint of the nearest line and becomes a new long line.
(19)$$\{\begin{array}{c} L=\sqrt{{\left(y_{0}-y_{1}\right)}^{2}+{\left(x_{0}-x_{1}\right)}^{2}}\\ \theta =\arctan\left|\frac{y_{0}-y_{1}}{x_{0}-x_{1}}\right| \end{array}.$$

In Figure 5, $\left(x_{0},y_{0}\right)$ and $\left(x_{1},y_{1}\right)$ are the coordinates of the two endpoints of nearest lines, and the length and direction angle of the connecting line are L and θ, respectively, connected when $L\leq L_{\mathrm{th}}\;\mathrm{and}\;\theta \leq \theta_{\mathrm{th}}$.

(3) Finally, screen out pseudo cracks (brick seams, light box edges, cables, steel plate edges, et al.) that have no obvious curve characteristics. So, the lines with obvious curve characteristics are retained, i.e. cracks.

The connected long lines are divided into *N* pieces, each piece contain at least 20 pixels, and the directional angle of each piece is calculated. The smaller the difference is, the closer it is to a straight line. i.e.:(20)$$\theta_{∆}=\theta_{1}-\theta_{i}=\arctan\left|\frac{y_{1}-y_{0}}{x_{1}-x_{0}}\right|-\arctan\left|\frac{y_{i}-y_{i-1}}{x_{i}-x_{i-1}}\right|$$
(21)$$\{\begin{array}{c} \theta_{\mathrm{AV}}=\frac{\sum_{i=1}^{N}|\theta_{∆}|}{N}<t,\;\;\;\;\;\;\;\mathrm{juged}\;\mathrm{as}\;\mathrm{pseudo}\;\mathrm{crack}\\ \theta_{\mathrm{AV}}=\frac{\sum_{i=1}^{N}|\theta_{∆}|}{N}\geq t,\;\;\;\;\;\;\;\;\;\;\;\;\mathrm{juged}\;\mathrm{as}\;\mathrm{crack} \end{array}\;$$

## 6. Steps of the Proposed Algorithm

The algorithm in this paper includes the DPG detection algorithm (Algorithm 1) for suspicious crack lines and the crack extraction algorithm (Figure 6) for breakpoint connections. The DPG algorithm is divided into the detection of vertical projection and detection of horizontal projection. Describing the vertical projection detection process by the DPG algorithm in pseudo-code is shown below. After calculating PV(x), $V_{1}$, $n_{Vlt}$, and $n_{Vrp}$ for the image I(x,y) using Equations (10), (12), and (14), respectively, they are used as input variables for the vertical projection detection algorithm.
**Algorithm 1: DPG algorithm (vertical projection)****Input:**I(x,y)***,***PV(x)***,***$V_{1}$***,***$\;n_{Vlt},\;n_{Vrp},\;k$***,***$\sigma_{1},\;\sigma_{2}$**Output: suspicious crack lines**1 [M,N]←size(I(x,y));
$2\;z\leftarrow \min\left(\left(\mathrm{PV}\left(0\right)+V_{1}\cdot n_{Vlt}\right),\left(\mathrm{PV}\left(M\right)+V_{1}\cdot n_{Vrp}\right)\right)$;
3 **for**
*x*←0 to M
4 $\;\;\;\;\;n_{l}\leftarrow \left[PV\left(x\right)-PV\left(0\right)\right]/V_{1}$
5 $\;\;\;\;\;n_{r}\leftarrow \left[PV\left(x\right)-PV\left(M\right)\right]/V_{1}$
6   **if** ($0\leq n_{l}\leq n_{Vlt}-1$)
7    **then**$\;\;\;T_{V}=PV\left(0\right)+V_{1}\cdot n_{l}-k\cdot V_{1}$;
8      **for**
$w\leftarrow \sigma_{1}\;to\;\sigma_{2}$
9       **if** (*T>*$T_{V}$)
10       **then** Marked as suspected crack line 1;
11     **end**
12    **end**
13   **else if** ($0\leq n_{r}\leq n_{Vrp}-1$)
14    **then**$\;\;\;T_{V}=PV\left(M\right)+V_{1}\cdot n_{r}-k\cdot V_{1}$;
15     **for**
$w\leftarrow \sigma_{1}\;to\;\sigma_{2}$
16       **if** (*T>*$T_{V}$)
17       **then** Marked as suspected crack line 2;
18      **end**
19     **end**
20  **else**
$21\;\;\;\;\;\;\;\;\;\;\;\;\;\;\;\;\;T_{V}=z-k\cdot V_{1}$;
22    **for**
$w=\sigma_{1}\;to\;\sigma_{2}$
23     **if**(*T>*$T_{V}$)
24      **then** Marked as suspected crack line 3;
25    **end**
26   **end**
27  **end**
28 **end**

Similarly, the DPG algorithm calculates the horizontal projection detection process using Equations (11), (13) and (15) and substitutes it into (17). The algorithm flow is similar to vertical projection.

Figure 6 is to combine the results of the DPG algorithm for the vertical projection and the horizontal projection detection, gets the suspicious crack lines as input, and extracts the cracks after the breakpoint connection. The flowchart is shown in Figure 6.

In the whole algorithm, there are several key parameters: *k,*
$\sigma_{1}$*,*
$\;\sigma_{2}$*,*
$\;T_{M}$*,*
$\;L_{th}$*,*
$\;\theta_{th}$*,* and  t. The values are given in the experimental section for the best estimates of the parameters.

## 7. Experiment and Discussion

### 7.1. Database Establishment and Algorithm Testing Platform

Verifying the algorithm in this paper was completed in the lining gallery captured on the site of the Tang-Ling-Shan Tunnel in Liaoning Province, China. The gallery was captured with a Nikon D5 camera, under the light of 330 W beam light, at the speed of 40 km/h, at a distance of about 1.5 m after adjusting the focal length, the shutter speed of the camera, and calculating the lens parameters. The resolution of the captured concrete lining image was 3721 × 5568.

### 7.2. Evaluation Index

In this paper, Precision and Recall are used to quantify the accuracy of crack detection.
(22)$$Precision=\frac{TP}{TP+FP}$$
(23)$$Recall=\frac{TP}{TP+FN}$$

Among them, *TP* (True Positive) is the number of cracks detected correctly, *FP* (False Positive) is the number of false detections, and *FN* (False Negative) is the number of missed detections. The larger the precision, the lower the false detection rate, and the larger the recall, the lower the missed detection rate.

### 7.3. Parameter Selection

The values of the key parameters *k*, $\sigma ,\;\mathrm{and}\;T_{M}$ in the algorithm in this paper are the best-estimated values got after analyzing the Precision, Recall, and Time curves through the experimental test shown in Figure 7.

In the multi-scale space fusion of *w* = 5−15 (σ ∊ (1.45, 4.33)), adjust the parameter *k* in Equation (16–17) with 0.1 as a step, the crack detection results obtained are shown in Figure 7a. It shows that with the increase of k, Recall increased, and Precision increased first, and then decreased. It can be seen from Equations (16–17) that the increase of k led to the decrease of threshold T. Therefore, from the Precision curve, because the image was dark on the whole, the edge information detected by the larger threshold was small. When *k* = 0.3, Precision is the largest, at this time Recall reaches 55%. As *k* continued to increase, *T* decreased, and the number of pseudo cracks detected increases. From the Recall curve, with the increase of *k*, more low-contrast edge information was detected, and more cracks and pseudo cracks were detected. When *k* ≥ 0.8, Recall could reach above 98% and then increase *k*—the value does not change much, but the time will increase a lot. The reason is that this threshold is close to the gradient threshold of the concrete pothole surface, resulting in much noise detection, which brings difficulties to the next crack identification. In conclusion, in order not to miss detection as far as possible, k was set at 0.8 as the best. 

When *k* = 0.8, adjust *w* = 3−20(σ = 0.87−5.78) and test the crack detection results with a single scale, respectively, as shown in Figure 7b. Where σ was calculated by substituting *w* into Equation (3), and *w* was set based on the statistics of crack and pseudo crack widths in 200 images, as shown in Table 1. It can be seen from the Recall curve that when *w* ∊ (5, 15), the crack detection effect was relatively good, and the detection effect was obvious at *w* = 5, 10, and 15, which was completely consistent with the crack width range in statistical Table 1. As can be seen from the time curve, when *w* = 4, 11, and 19, the time consumption significantly increased, because the corresponding pseudo crack information of concrete pits, brick seams, and light box edges in Figure 1 were detected in large quantities. Besides, when *k* = 0.8, observe the Recall value of a single scale in Figure 7b and multi-scale in Figure 7a. It can be seen that the maximum value of a single scale was less than 60%, while the maximum value of multi-scale can reach more than 98%, i.e., the multi-scale detection of cracks makes the missed detection rate significantly reduced. Besides, when *w* = 15, Precision is the highest, indicating that the main width of the crack in this Figure occupies 15 pixels. So, σ1 = 1.45(*w* = 5), σ2 = 4.33(*w* = 15) are the best.

In Figure 7b, although when *k* = 0.8 and σ ∊ (1.45, 4.33), the Recall is very high, but Precision is less than 10%, because a large amount of pseudo crack information can be detected under the condition of ensuring no omission. Through Equation (18), the maximum diameter length of the detected line is screened. Adjust $T_{m}\;$= 20−300, and test the crack screening results respectively, as shown in Figure 7c. It can be seen that the Precision significantly increased. After $T_{m}\;$≥ 60, Precision has exceeded 10%. When $T_{m}\;$= 180, Precision reached the maximum, and Recall decreased significantly. After$\;T_{m}\;$≥ 220, Recall changed little. To retain as many cracks as possible and screen out as many pseudo cracks as possible, in the rough screening, $T_{m}$ is preferably 180.

At the time of a broken connection, the main parameters are the straight-line distance $L_{th}$, the direction $\theta_{th}$ from the endpoints of the nearest two broken lines, and the threshold t for determining the degree of curve. After observing the rough screening results in Figure 8, there are intermittent and short distances in the middle of the crack line (such as in the red oval frame), and the lines and lines at the pseudo crack are far away (such as in the red, dashed box), or the nearest neighbor line has an obtuse angle change in the direction (such as in the red solid line frame), so it was observed that $L_{th}=$300 and $\theta_{th}=\;\pi /2$. Also, the difference in direction angles between straight lines is small, here t = π/18.

According to the algorithm flow, the results of the key steps of the detection algorithm in this paper are shown in Figure 8.

### 7.4. Crack Detection with Different Projection Curves Distributions

Since the gray values range of the projection curve distribution in Figure 1 is among 7.5−33.3 in the tunnel lining gallery, randomly select the two images with an obvious difference in the gray values range of the projection curve distribution from Figure 1. The detection results using the algorithm in this paper are shown in Figure 9 and Figure 10. 

The gray values range of the projection curves distribution in Figure 8 and Figure 9 was 10.4−50.6 and 2.5−16, respectively, showing as one lighter and one darker. The algorithm used in this paper achieved good results in detecting cracks in both images.

### 7.5. Comparison with Other Methods

The detection algorithm mentioned in Reference [18] includes two parts: one is crack detection based on multi-scale ridge edges and the other is pseudo crack removal and crack connection. Algorithm 1 includes three parts: light balanced processing, single-scale detection, and multi-scale fusion. Images 1 and 2 in this library were detected, respectively, and the results are shown in Figure 11 and Figure 12. Two hundred randomly selected images with cracks in the gallery for detection were chosen for a quantitative comparison with the detection results of the algorithm in this paper, as shown in Table 2.

According to the detection results in Reference [18], although uniform light processing was done, the inconsistencies between the cracks and the surrounding background were not taken into account. As a result, the same threshold was used to detect the cracks at the place with a small difference (both sides of the image), while a large number of noises were detected at the place with a large difference (the middle of the image), resulting in a small Recall value. Besides, only four scales were selected in Reference [18], which resulted in a certain degree of omission, resulting in a small Precision value. The pseudo cracks elimination in Reference [18] was accomplished through the length, width, macroscopic continuity and consistency of adjacent cracks which can remove scattered pseudo cracks well. However, there were other linear pseudo-cracks with the above characteristics in this gallery, so they are not well removed. 

Also, because the difference between cracks and surrounding background in this library is weak, the continuity of the detected lines is poor, multiple small segments in the coarse screening have been removed, and this article relied on manual annotation sample comparison tests. Manual annotation was done according to crack macroscopic to draw lines, and the real cracks on the lining surface formed themselves was discontinuous, so the manual annotation on the number of pixels was greater than the real cracks, making the Recall in the result of the algorithm smaller, but with the same manually annotation samples to compare the two algorithms, the difference was still meaningful.

It can be seen from the comparison that the setting of dynamic threshold in this paper makes it possible to extract cracks without the influence of uneven illumination, and to a certain extent, the rate of missed detection and false detection was reduced.

## 8. Conclusions

This paper proposed a dynamic partitioned Gaussian crack detection algorithm based on the distribution of projection curves. In the new DPG model proposed, regions are divided by image projection curve, and the center point of the crack line is detected by setting dynamic local threshold *T*, and the crack lines of different widths were detected by setting different scale factor σ. Later in the engineering example, which had no obvious differences in brightness and contrast between the crack pixels and the background pixels, through Recall, Precision, and Time curve to estimate the optimal parameters of the algorithm, and from qualitative and quantitative analysis, it verified the algorithm in this paper, that it can effectively eliminate the influence of uneven illumination image detection of cracks, and the Recall of the crack detection can reach above 96%. Finally, the crack breakpoint connection algorithm and crack morphology were combined to screen the cracks, which increased Precision by more than 70%. Therefore, this paper provides a new idea for detecting lines in uneven illumination images. 

However, there are still some problems, i.e., when the brightness and contrast difference between the crack and the surrounding background pixel is small, it is necessary to set a low threshold for detection, resulting in a large number of pseudo cracks and small real cracks. However, in the process of coarse screening, it is easy to delete pseudo cracks and delete real cracks by mistake, so that although before the rough screening, the Recall is more than 96% after the screening is significantly reduced. Therefore, the next step is how to improve the effectiveness of the crack screening algorithm.

## Figures and Tables

**Figure 1 sensors-20-03973-f001:**
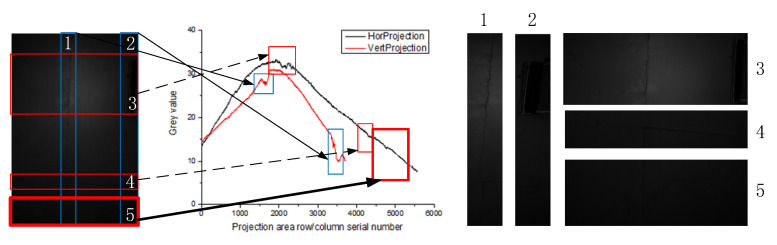
Tunnel lining image and its horizontal (vertical) gray projection curve.

**Figure 2 sensors-20-03973-f002:**
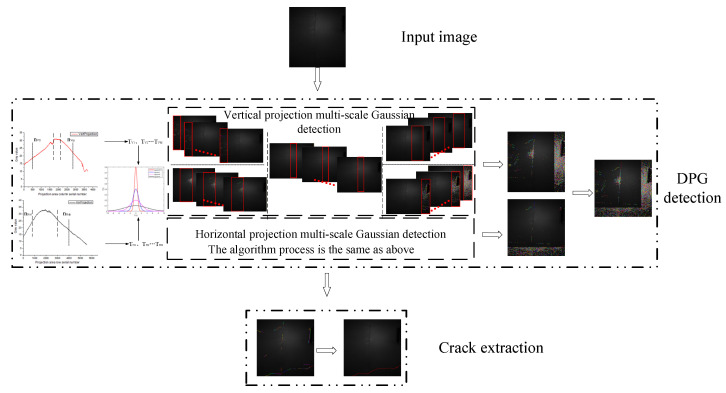
The framework of the algorithm in this paper: input image, dynamic partition Gaussian (DPG) detection, crack extraction.

**Figure 3 sensors-20-03973-f003:**
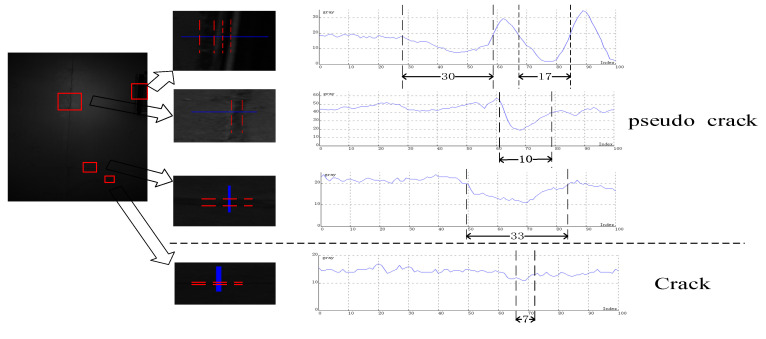
Lines of various widths in the tunnel lining image and its profile gray curve.

**Figure 4 sensors-20-03973-f004:**
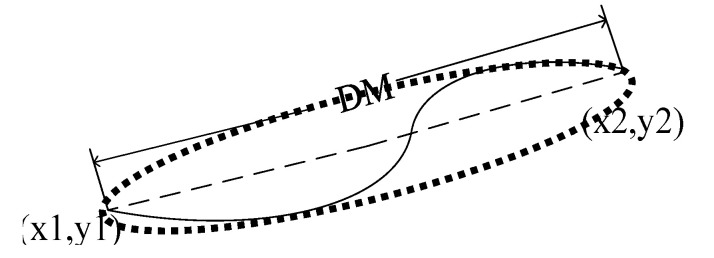
Maximum diameter of the circumscribed ellipse of the line.

**Figure 5 sensors-20-03973-f005:**
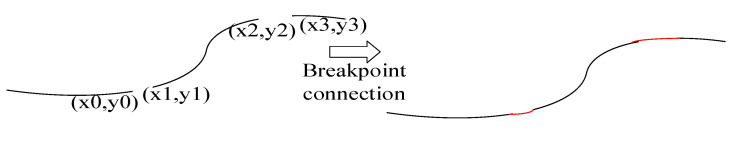
Breakpoint connection process.

**Figure 6 sensors-20-03973-f006:**
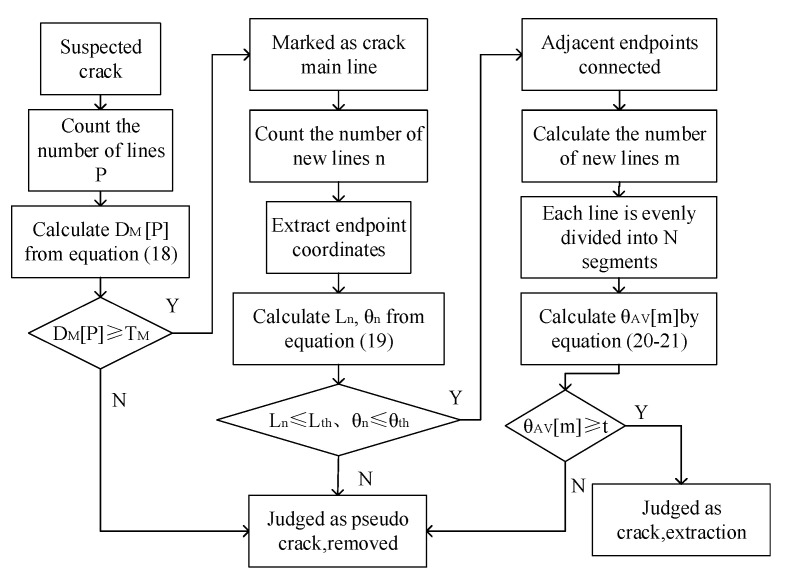
Crack extraction flow chart.

**Figure 7 sensors-20-03973-f007:**
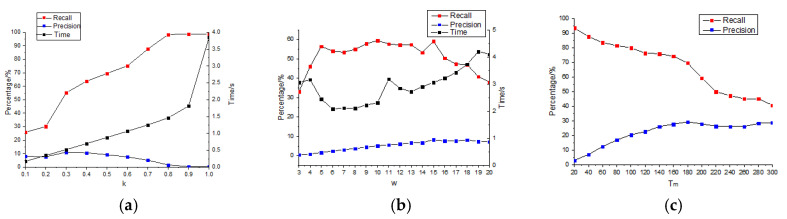
The crack detection results after adjusting: (**a**) *k*, (**b**) w(σ); (**c**) $T_{m}$.

**Figure 8 sensors-20-03973-f008:**
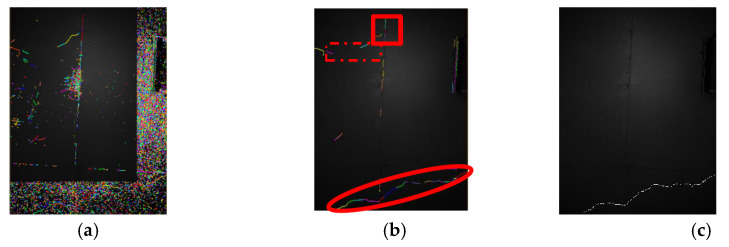
The crack detection result of image 2 by the algorithm in this paper: (**a**) DPG detection results; (**b**) coarse screening results; (**c**) crack extraction results after broken connection.

**Figure 9 sensors-20-03973-f009:**
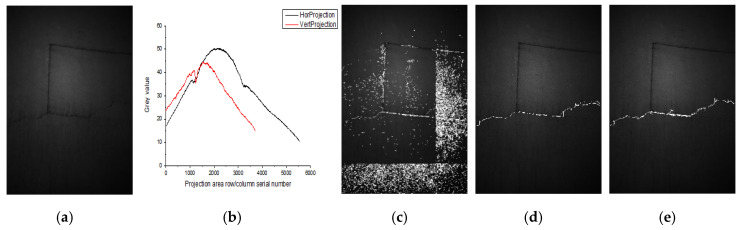
The crack detection result of image 2 by the algorithm in this paper (**a**) image 2; (**b**) gray projection curve of image 2; (**c**) DPG detection result; (**d**) crack extraction results; (**e**) manual annotation.

**Figure 10 sensors-20-03973-f010:**
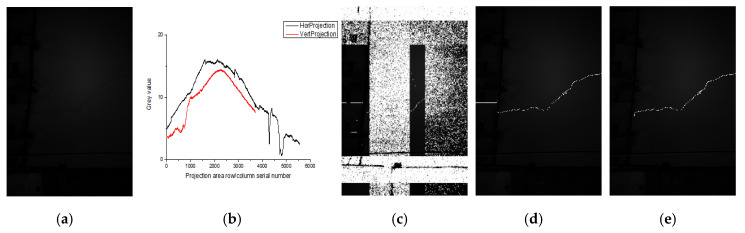
The crack detection result of image 3 by the algorithm in this paper: (**a**) image 3; (**b**) gray projection curve of image 3; (**c**) DPG detection result; (**d**) crack extraction results; (**e**) manual annotation.

**Figure 11 sensors-20-03973-f011:**
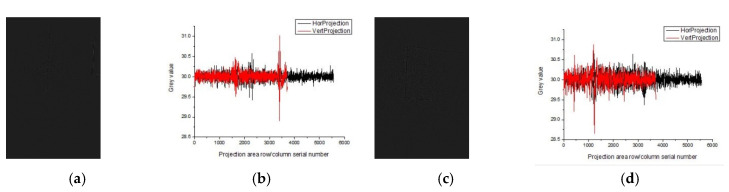
Detection results of step 1 of the crack detection algorithm in Reference [8]: (**a**) Light balanced result of image 1; (**b**) gray projection curve of light balanced image1; (**c**) light balanced result of image 2; (**d**) gray projection curve of light balanced image2.

**Figure 12 sensors-20-03973-f012:**
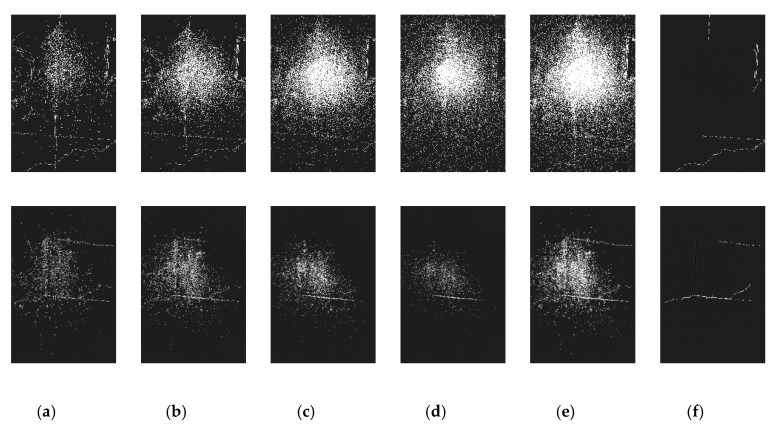
Detection results of steps 2 and 3 of the crack detection algorithm in Reference [18]: (**a**) detection results of step 2(σ = 8); (**b**) detection results of step 2(σ = 4); (**c**) detection results of step 2(σ = 2); (**d**) detection results of step 2(σ = 1); (**e**) detection results of step 2(multiscale fusion); (**f**) detection results of step 3.

**Table 1 sensors-20-03973-t001:** Statistical table of crack and pseudo crack width.

Type	*w*(pixels)	*σ*
Concrete potholes	< 5	< 1.45
Cracks	5–15	1.45–4.33
Other pseudo cracks	≥ 10	≥ 2.89

**Table 2 sensors-20-03973-t002:** Algorithm comparison results.

Algorithm	Recall after Detection	Precision after Detection	Recall after the Crack Screening	Precision after the Crack Screening
[18]	94.2%	2%	39%	50.2%
In this paper	96.4%	2.3%	40%	78.8%

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
