# Peer review of "Dynamic Partition Gaussian Crack Detection Algorithm Based on Projection Curve Distribution"

_sensors, 2020, doi:10.3390/s20143973_

Round 1
Reviewer 1 Report
The paper presents an algorithm for high precision cracks detection, in the cases of uneven light, with application on a lining tunnel case study. The paper is interesting and well written, easy to follow and formally precise. For these reasons, I recommended it for the publication in "Sensors" Journal. Nevertheless, I suggest to authors of taking into account some minor suggestions, as followed listed:
- At row 41, authors highlighted the necessity of capturing cracks in some civil infrastructures. In my opinion, this procedure could be used overall for simple structures, such as the existing buildings that suffer of concrete decay. The phenomenon could be very important especially in the vulnerability analysis of existing structures, accounting for all the phases of material characterization (Some information could be find in "Seismic assessment of irregular existing building: Appraisal of the influence of compressive strength variation by means of nonlinear conventional and multimodal static analysis" ) and subsequent modelling hypotheses (Some information could be find in "A practical approach for estimating the floor deformability in existing RC buildings: evaluation of the effects in the structural response and seismic fragility "). If the authors believe relevant this aspect, they could review this part of the Introduction.
- Row 77 - sett --> set
- How did the authors defined Figure 1? Maybe a reference or an explanation of Projection curve is suggested
- Row 117 - i.e. --> For example
- The paragraph at Row 150 is not very clear. Maybe authors could be rewrite it
- The title of Section 6 is not very formal. It should be changed
- Row 260 - the paragraph is not very clear. Maybe it could be better explained.
- Figure 11 presents captions in Chinese. Please, to replace them
- Observing the good capacity of the algorithm to survey little cracks, I'm guessing what is the main advantage in this accurate survey, also considering that the concrete cracks in this kind of infrastructure is natural, due to phenomena of viscosity and shrinkage?
Reviewer 2 Report
Authors address an important issue and explore an image processing based automated solution. Road and specially highway maintenance require constant surveillance and quality monitoring. Advances in image aquisition digitalisation and processing hardware and software evolved very positively over last two decades and can now be employed with fair to good sucess in several quality control tasks specially when speed and low cost are key issues. In this paper authors address the fast crack detection and inspection of the tunnel lining in order to allow the timely repair and so the safety of the tunnels. A Dynamic Partitioned Gaussian (DPG) algorithm on projection curve distribution was developed in order to cope with the efficient detection of cracks in real conditions of uneven illumination varying brightness and contrast with partitionned backgrounds. Different crack widths and lenghts are considered and well as brook lines connection. The algorithm is basic but sound enough. Neverthless further work is advised specially in trying to overcome the limited contrast bettween the background and the crack. I sugest considering bi- or multi- wavelenght illumination. Even with just using two illumination collour may reduced the number of faulty crack detection significantly rendering the process to be truly effective in a practical application stand point. In graphs such as the one on figure 7 avoid connecting data points and try to adjust a suitable trend curve to the data points.
